

# On the atmospheric budget of ethylene dichloride and its impact on stratospheric chlorine and ozone (2002-2020)

Ryan Hossaini[1], David Sherry[2], Zihao Wang[3,4], Martyn P. Chipperfield[3,5], Wuhu Feng[3,6], David E. Oram[7,8], Karina E. Adcock[8], Stephen A. Montzka[9], Isobel J. Simpson[10], Andrea Mazzeo[1], Amber A. Leeson[1], Elliot Atlas[11], and Charles C.-K Chou[12].

[1]Lancaster Environment Centre, Lancaster University, Lancaster, UK.
[2]Nolan Sherry and Associates (NSA), London, UK.
[3]School of Earth and Environment, University of Leeds, Leeds, UK.
[4]Department of Ocean Sciences and Engineering, Southern University of Science and Technology, Shenzhen, China.
[5]National Centre for Earth Observation, University of Leeds, Leeds, UK.
[6]National Centre for Atmospheric Science, University of Leeds, Leeds, UK.
[7]National Centre for Atmospheric Science, University of East Anglia, Norwich, UK.
[8]School of Environmental Sciences, University of East Anglia, Norwich, UK.
[9]NOAA Global Monitoring Laboratory (GML), Boulder, CO, USA.
[10]Department of Chemistry, University of California-Irvine, Irvine, CA, USA.
[11]Department of Atmospheric Sciences, RSMAS, University of Miami, Miami, Florida, USA.
[12]Research Center for Environmental Changes, Academia Sinica, Taipei, Taiwan.

*Correspondence to*: Ryan Hossaini (r.hossaini@lancaster.ac.uk)

**Abstract.** Ethylene dichloride (EDC), or 1-2-dichloroethane, is an industrial very short-lived substance (VSLS) whose major use is as a feedstock in the production chain of polyvinyl chloride (PVC). Like other chlorinated VSLS, transport of EDC (or its atmospheric oxidation products) to the stratosphere could contribute to ozone depletion there. However, despite annual production volumes greatly exceeding those of more prominent VSLS (e.g. dichloromethane), global EDC observations are sparse, thus the magnitude and distribution of EDC emissions and trends in its atmospheric abundance are poorly known. In this study we performed an exploratory analysis of the global EDC budget between 2002 and 2020. Combining bottom-up data on annual production and assumptions around fugitive losses during production and feedstock use, we assessed the EDC source strength required to reproduce atmospheric EDC observations. We show that the TOMCAT/SLIMCAT 3-D chemical transport model (CTM) reproduces EDC measurements from various aircraft missions well, including HIPPO (2009-2011), ATom (2016-2018) and KORUS-AQ (2016), along with surface measurements from South East Asia, when assuming a regionally varying production emission factor in the range 0.5-1.5%. Our findings imply substantial fugitive losses of EDC and/or substantial emissive applications (e.g. solvent use) that are poorly reported. We estimate EDC's global source increased by ~45% between 2002 (349±61 Gg/yr) and 2020 (505±90 Gg/yr) with its contribution to stratospheric chlorine increasing from 8.2 (±1.5) ppt Cl to ~12.9 (±2.4) ppt Cl over this period. EDC's relatively short overall tropospheric lifetime (~83 days) limits, though does not preclude, its transport to the stratosphere and we show that its impact on ozone is small at present. Annually averaged, EDC is estimated to have decreased ozone in the lower stratosphere by up to several ppb (<1%) in 2020, though a larger effect in the springtime Southern Hemisphere polar lower stratosphere is apparent (decreases of up





to ~1.3%). Given strong potential for growth in EDC production tied to demand for PVC, ongoing measurements would be of benefit to monitor potential future increases in its atmospheric abundance and its contribution to ozone depletion.

## 1 Introduction

Very short-lived substances (VSLS) are a class of halogenated chemicals with local surface lifetimes typically less than ~6 months, leading to spatial and temporal heterogeneity in their tropospheric abundance (e.g. WMO, 2018, 2022). Despite short lifetimes relative to long-lived ozone-depleting substances (ODSs) controlled by the Montreal Protocol, such as chlorofluorocarbons (CFCs) and hydrofluorocarbons (HCFCs), a range of both natural and anthropogenic VSLS have been detected in the lower stratosphere (e.g. Laube et al., 2008; Hossaini et al., 2019; Keber et al., 2020). This has motivated research into the possible impacts of VSLS on stratospheric ozone and ozone trends (e.g. Salawitch et al., 2005; Feng et al., 2007; Falk et al. 2017; Bednarz et al., 2022, 2023; Villamayor et al., 2023). The most prominent VSLS with significant industrial sources are chlorinated compounds (Cl-VSLS), including dichloromethane ($CH_2Cl_2$) and chloroform ($CHCl_3$). These gases have a non-zero ozone depletion potential (ODP, Claxton et al., 2019) and global emissions of both have increased considerably in recent years, particularly from Asia (e.g. Hossaini et al., 2017; Fang et al., 2019; Say et al., 2019; Claxton et al., 2020; An et al., 2021, 2023).

The molecule 1,2-dichloroethane ($CH_2ClCH_2Cl$), known commonly as ethylene dichloride (EDC), is a further chlorinated VSLS, produced industrially in large volumes worldwide. In the USA, for instance, some ~9,000-14,000 Gg of EDC are estimated to have been produced annually in the period 2011 to 2015 (ATSDR, 2022) and global total production capacity in 2020 was estimated at ~60,000 Gg (TEAP, 2022). EDC's main use is as a chemical intermediate in the manufacture of vinyl chloride monomer (VCM), a raw material in the production of the widely used plastic, polyvinyl chloride (PVC). Over 95% of EDC consumption is estimated to be in VCM production (UNEP, 2002; ECHA, 2012; CEH, 2023) which, in principle, is a largely non-emissive application (i.e. because EDC is consumed in reaction). Like other halocarbons however, fugitive release of EDC to the atmosphere may occur during its production, storage and transportation (TEAP, 2022). Other known but relatively minor uses of EDC include: (1) in the production of other chemicals, such as ethyleneamines (e.g. Ayres and Ayres, 1997), (2) historically, as a lead scavenger in fuels (e.g. Falta et al., 2005), and (3) in various applications on account of being an effective solvent, such as metal degreasing (EPA, 2020), and in organic and medicinal chemistry (e.g. Jordon et al., 2021). Due to concern over its toxicity, regulatory controls restricting commercial EDC uses are in place in some regions, including the European Union (Sherwood et al., 2018) where EDC was placed in Annex XIV of the EU's REACH (Registration, Evaluation, Authorisation and Restriction of Chemicals) regulation in 2016.

In contrast to other major chlorinated VSLS (e.g. $CH_2Cl_2$, $CHCl_3$, $C_2Cl_4$), the National Oceanic and Atmospheric Administration (NOAA) and Advanced Global Atmospheric Gases Experiment (AGAGE) global monitoring networks do





not yet routinely report surface EDC measurements and there are no other archived long-term observational records. In consequence, the global EDC budget and trends in its atmospheric abundance are poorly known. The current paucity of
global EDC surface measurements also prevents the assessment of its global source using top-down inverse methods, as performed for other industrial VSLS (Claxton et al., 2020). Measurements of EDC from a limited number of aircraft campaigns in various world regions indicate typical Northern Hemisphere (NH) boundary layer mole fractions in the range ~10-20 ppt (Engel and Rigby et al., 2018; Roozitalab et al., 2024). However, far larger levels have also been detected in East and South-East Asia, including mole fractions >1 ppb in China at reportedly both urban and background sites (Lyu et al.
2020 & references therein). Based on air samples obtained from surface sites in Taiwan and Malaysia in 2013 and 2014, Oram et al (2017) reported median EDC mole fractions of 85.4 (16.7–309) ppt and 21.7 (16.4–120) ppt, respectively, with a strong correlation of EDC with $CH_2Cl_2$ observed at both sites. Combining this relationship with a bottom-up estimate of regional $CH_2Cl_2$ emissions, the same study inferred Chinese EDC emissions to be of the order of 203 (±9) Gg/yr.

Based on a combination of high-altitude aircraft observations and modelling, Cl-VSLS were estimated to provide ~130 (100-160) ppt Cl to the stratosphere in 2019 (Laube and Tegtmeier et al., 2022). Although this represents just ~4% of total stratospheric chlorine (which principally is from long-lived ODSs that are now controlled by the Montreal Protocol), increasing VSLS amounts have slowed the rate at which chlorine is decreasing in the stratosphere (Hossaini et al., 2019; Bednarz et al., 2022). Additionally, far larger local injections of Cl-VSLS (including EDC) into the Northern Hemisphere
(NH) extratropical lower stratosphere (LS) have been reported (Adcock et al., 2021; Lauther et al., 2022), reflecting transport via the Asian summer monsoon anticyclone and the co-location of relatively strong Asian emissions with efficient vertical ascent (e.g. Randel et al., 2010). While $CH_2Cl_2$ remains the largest contributor to stratospheric chlorine from VSLS (Laube and Tegtmeier et al., 2022), the large volumes of EDC produced worldwide, its substantial global trade, and the potential for future growth tied to PVC demand (e.g. in the building and construction industries) means it is of interest to establish EDC's
present-day atmospheric budget and fate.

In this study, we have analysed global EDC production data between 2002 and 2020 and used it to create a set of gridded global emissions for different assumed emission factors describing fugitive EDC losses. Using the TOMCAT/SLIMCAT 3-D chemical transport model (CTM), we evaluated the realism of these emissions by assessing the model's ability to
reproduce various aircraft measurements of EDC, thereby providing new constraints on its global source. The CTM was used to quantify the likely contribution of EDC and its products to stratospheric chlorine and thus the potential impact of EDC emissions on stratospheric ozone. The paper is structured as follows. Section 2 describes our approach to creating the EDC emission inventories, as well as the CTM, the simulations performed, and observational datasets used. Our results are presented in Section 3, including on the inferred magnitude of global EDC emissions (Section 3.1), EDC's budget and
contribution to stratospheric chlorine (Section 3.2), and EDC's impact on stratospheric ozone (Section 3.3). A summary of key findings and concluding remarks is given in Section 4.





## 2. Data, Methods and Model

### 2.1 Bottom-up data on EDC production

EDC is manufactured industrially via the direct chlorination of ethene or via its oxychlorination with hydrogen chloride.
Estimated annual EDC production data were compiled biennially by Nolan Sherry Associates (NSA) over the period 2002 to 2020 (**Table 1**). Data from NSA were reported in the most recent Technology and Economic Assessment Panel (TEAP) report to the parties of the Montreal Protocol (TEAP, 2022) and have been utilised in a range of recent scientific papers (e.g. Chipperfield et al., 2018; Claxton et al., 2020). Analysis by NSA makes use of their extensive database of halocarbon production and production capacities, industry data, and public reports, and is refined through industry dialogue. NSA's
analysis of EDC production includes assessment of downstream products (VCM and PVC), accounting for several specific industry and market factors and trade movements. This includes the fact that VCM production may not always occur via the "ethylene route", which uses EDC at a rolling ratio of 1.6 units EDC to VCM, but the "acetylene route", which involves the direct production of VCM from acetylene's reaction with hydrogen chloride (i.e. no EDC involved). The latter approach is prevalent in, for example, China, meaning that Chinese EDC production is relatively modest compared to other major global
economies. Note that at the country level, some of the data available to NSA are proprietary in nature and confidential. On this basis and to aid the discussion and presentation, data have been aggregated into 13 broader geographical regions for which we discuss production and emissions. These regions cover all the world's major industrialised zones and their boundaries (**Figure 1**) are based on the region definitions used in Phase 2 of the Hemispheric Transport of Air Pollution (HTAP) project (e.g. Huang et al., 2017).

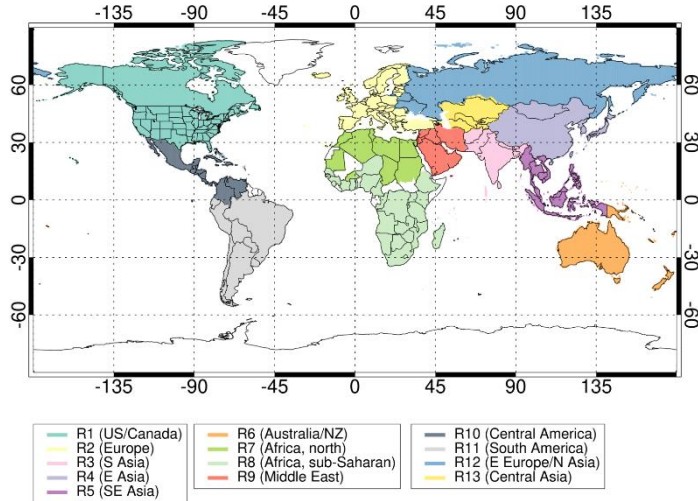


**Figure 1.** *Definitions of the 13 geographical regions considered for EDC production and emissions*





Evident from the data in **Table 1** is that EDC is produced in large quantities (~52,000 Gg in 2020) and that global production increased by ~29% between 2002 and 2020. North America, Europe, and S, E and SE Asia (regions 1-5) are estimated to account for ~86% of world production in 2020. Noting that the end-product of EDC's principal industrial use (i.e. PVC) is closely followed and reported on by both business performance analysts and environmentalists, the production data from NSA is estimated to be accurate to within around ±5%. Although estimates of EDC production in peer-reviewed literature are scarce, some independent figures exist with which to compare. For instance, data cited by the U.S. Environmental Protection Agency (EPA) places the annual volume of EDC produced in the USA at 12,750 Gg in 2011 (EPA, 2020). Interpolating between years on either side (recalling that production data from NSA was provided biennially), the corresponding U.S. production from NSA is 13,145 Gg (2011), i.e. in close agreement with U.S. EPA figure (within ~3%). An assessment around health aspects of EDC exposure placed European production at more than 10,000 Gg per annum (Cherrie et al., 2011), consistent with the NSA data in **Table 1**. Few estimates of EDC production in Asia exist in the peer-reviewed literature. However, an estimate of Chinese EDC production in the year 2010 of 2,708 Gg (Chinabaogao, 2012) is very similar to that for China from NSA in the same year (2,700 Gg).

*Table 1. Estimated annual production of EDC (Gg) from NSA in the 13 world regions in Figure 1.*

| R# | Region name | 2002 | 2004 | 2006 | 2008 | 2010 | 2012 | 2014 | 2016 | 2018 | 2020 |
|---|---|---|---|---|---|---|---|---|---|---|---|
| 1 | US/Can | 14429 | 14929 | 13894 | 12649 | 12789 | 13499 | 14199 | 16199 | 18199 | 16749 |
| 2 | Europe | 11176 | 11757 | 12055 | 11300 | 11680 | 11771 | 11739 | 11799 | 11749 | 11499 |
| 3 | S Asia | 284 | 257 | 239 | 274 | 449 | 479 | 399 | 423 | 457 | 444 |
| 4 | E Asia | 8951 | 9545 | 10563 | 10642 | 10771 | 10534 | 10723 | 11738 | 13889 | 14147 |
| 5 | SE Asia | 1114 | 1249 | 1239 | 1409 | 1549 | 1759 | 1799 | 2014 | 2153 | 2199 |
| 6 | Aus/NZ | 0 | 0 | 0 | 0 | 0 | 0 | 0 | 0 | 0 | 0 |
| 7 | Afr. N | 117 | 144 | 144 | 139 | 137 | 224 | 314 | 314 | 339 | 555 |
| 8 | Afr. Sub-S | 207 | 175 | 207 | 151 | 263 | 223 | 263 | 271 | 263 | 239 |
| 9 | Mid. East | 1636 | 1830 | 2076 | 1817 | 2552 | 2807 | 2924 | 3339 | 3436 | 3327 |
| 10 | Cen. Am. | 394 | 279 | 524 | 579 | 639 | 629 | 609 | 89 | 89 | 89 |
| 11 | S Am. | 1129 | 1234 | 1379 | 1639 | 1559 | 1429 | 1579 | 1609 | 1389 | 999 |
| 12 | E Eur/N Asia | 1011 | 976 | 1294 | 1274 | 1214 | 1034 | 1179 | 1359 | 1819 | 1889 |
| 13 | Cen. Asia. | 0 | 0 | 0 | 0 | 0 | 0 | 0 | 0 | 0 | 0 |
|  | **Global total** | **40457** | **42384** | **43623** | **41882** | **43611** | **44398** | **45737** | **49164** | **53792** | **52146** |



*Table 2. As Table 1 but for consumption.*

| R# | Region name | 2002 | 2004 | 2006 | 2008 | 2010 | 2012 | 2014 | 2016 | 2018 | 2020 |
|---|---|---|---|---|---|---|---|---|---|---|---|
| 1 | US/Can | 13024 | 13503 | 12827 | 11826 | 12103 | 12744 | 12955 | 14867 | 16882 | 15252 |
| 2 | Europe | 10980 | 11632 | 11975 | 11114 | 11354 | 11463 | 11539 | 11503 | 11646 | 11327 |
| 3 | S Asia | 579 | 526 | 540 | 569 | 798 | 1016 | 993 | 1066 | 1307 | 1186 |
| 4 | E Asia | 10718 | 11510 | 11920 | 11712 | 11836 | 11373 | 11960 | 12935 | 14642 | 14595 |
| 5 | SE Asia | 1365 | 1423 | 1498 | 1742 | 1848 | 1967 | 1834 | 2339 | 2492 | 2545 |
| 6 | Aus/NZ | <1 | <1 | <1 | <1 | <1 | <1 | <1 | <1 | <1 | <1 |
| 7 | Afr. N | 118 | 145 | 145 | 140 | 147 | 260 | 448 | 482 | 536 | 1011 |
| 8 | Afr. Sub-S | 202 | 175 | 208 | 152 | 264 | 224 | 264 | 281 | 264 | 240 |
| 9 | Mid. East | 1095 | 1234 | 1458 | 1264 | 1990 | 2265 | 2386 | 2627 | 2614 | 2603 |
| 10 | Cen. Am. | 390 | 284 | 535 | 568 | 662 | 631 | 610 | 91 | 106 | 90 |
| 11 | S Am. | 1036 | 1008 | 1223 | 1580 | 1395 | 1415 | 1563 | 1610 | 1479 | 1403 |
| 12 | E Eur/N Asia | 944 | 938 | 1288 | 1246 | 1210 | 1034 | 1179 | 1360 | 1820 | 1890 |
| 13 | Cen. Asia. | <1 | <1 | <1 | <1 | <1 | <1 | <1 | <1 | <1 | <1 |
| | **Global total** | **40457** | **42384** | **43623** | **41882** | **43611** | **44398** | **45737** | **49164** | **53792** | **52146** |


The demand for EDC in both producing and non-producing countries was evaluated from trade data. Net imports (gross imports minus gross exports) were calculated for a total of ~150 countries over our study period (2002-2020) using publicly available trade statistics accessed via the online UN Comtrade Database (https://comtradeplus.un.org/). Assuming global imports should equal global exports in a given year, net imports should sum to zero across the globe. However, due to

imperfections in reported trade data (known to afflict many commodities besides EDC) this was found not to be the case. Although the imbalance was small compared to the large production volumes of EDC, we elected to reconcile the trade data using the method of Zou et al. (2023). Briefly, where a record of EDC trade is recorded by the importer but not the exporter (or vice versa), the missing trade is filled in. Where records match but the trade quantities differ, the larger of the two was adopted (Zou et al., 2023). This approach balances global EDC trade and prevents errors in trade statistics from confounding

our subsequent analysis.

## 2.2 EDC emissions

Emissions of EDC may in principle arise during its (1) production, (2) use as a feedstock, (3) transportation, and (4) any emissive uses (e.g. as a solvent). Items 1-3 represent fugitive emissions that may arise from, for example, the operation and maintenance of chemical plants, along with bulk storage and other industrial processes where unintended leakage can occur.

In a fully explicit bottom-up inventory, production emissions may be calculated as the product of annual EDC production




and a suitable emission factor. Similarly, feedstock use emissions, which are additional and additive, may be calculated from the quantity of EDC used as feedstock and a further emission factor (e.g. TEAP, 2022). However, although EDC is principally used as a feedstock in the manufacture of VCM, with some assessments placing this use at >98% (CEH, 2023), the precise quantity and how this may have varied over time and across regions is unknown. Note, even if 98% of EDC use is in producing VCM, this does not imply that the remaining 2% is used in emissive applications. This is because EDC also finds use as an intermediate in the production of other chemicals, including ethyleneamines and other chlorinated solvents (Section 1; TEAP, 2022). Analysis by NSA suggests that these two sectors contribute of the order 600-800 Gg/yr of EDC feedstock use. In our idealised framework for calculating EDC emissions we assume that 100% of EDC use (consumption) is as feedstock.

The annual total EDC emission per country in year $t$ was calculated using **Equation 1**.

$$Emission(t) = P(t)\alpha_1 + C(t)\alpha_2 + I(t)\alpha_3 \qquad (1)$$

The first term on the right denotes production emissions calculated from the time-varying production (P) data provided by NSA. There are 36 producing countries in the NSA database to which this term applies. The second term on the right denotes feedstock use emissions calculated based on consumption (C) data. Consumption (production + net imports) was calculated for ~150 countries in total. The EDC-producing countries dominate global consumption, with non-producers accounting for less than 0.6% of the global total. Production and consumption data in **Tables 1** and **2** are aggregated regional totals obtained from country-level analysis. The third term on the right of Equation 1 represents fugitive emissions during supply chain. We have elected to apply these in the country of import and they are calculated from gross import (I) data.

In Equation 1, emission factors for production, feedstock use, and supply chain emissions are denoted by $\alpha_1$, $\alpha_2$, and $\alpha_3$, respectively. Tight emission controls throughout the whole EDC production cycle and its supply chain, especially in more developed countries, are expected to occur to minimise loss of useful material, to control costs in an extremely competitive industry, and also for possible legal compliance. For EDC, all α values in developed countries are thus expected to be small and likely to lie towards the lower end of the plausible ranges reported in the literature for other gases (TEAP, 2022). For context, fugitive emissions from production of other halocarbons have typically been estimated at ~0.5% on production (IPCC/TEAP, 2005). However, due to differences in plant operations and regulatory requirements in different world regions, regional differences in emission factors are likely. For EDC, we examined a range of emission factors around the above value, varying $\alpha_1$ between 0.1% through to 0.6% for developed countries. For developing countries, approximated as those operating under Article 5 (A5) of the Montreal Protocol, we assume a multiplier of 3. The different scenarios are labelled according to their developed country $\alpha_1$ emission factor (see **Table 3**). Fugitive emissions from feedstock uses are generally expected to be lower than those from production (TEAP, 2022). We assumed a fixed feedstock emission factor of $\alpha_2 = 0.1\%$, representing the "low" estimate reported by TEAP (2022). Note, where consumption is negative (i.e. exports exceed





production plus imports), we assume no feedstock use emission. Similarly, we adopt a fixed supply chain emissions factor of $\alpha_3 = 0.1\%$, representing the "low" estimate for distribution emissions reported by TEAP (2022). The resulting estimated range of total fugitive emissions is 146-594 Gg/yr in 2020 (**Table 3**).

Many modern EDC-producing plants are integrated on-site with VCM/PVC production (Cherrie et al., 2011). If the processes are seamless, the distinction between fugitive losses from production and fugitive losses from feedstock use may be less clear cut compared with other gases. An alternative framework (not adopted) thus might be to consider a single emission factor, applied to production, that encapsulates all possible leakage over EDC's internal lifetime within a plant. As we elected to consider global trade, and hence use consumption in conjunction with production, it was necessary to treat the two terms separately. We note that our overarching goal is to examine the impact of EDC on stratospheric ozone using a global model calibrated to reproduce tropospheric observations of EDC. The overall magnitude and location of EDC emissions is thus important, but the detail of the fugitive source is secondary.

For inclusion in the CTM, the calculated biennial EDC emissions (**Table 3**) were linearly interpolated to give annual records over the 19-year study period (2002 to 2020). The emissions were aggregated onto a global 0.5°×0.5° grid using the country mask of Perrette (2023). This mask was developed for the Inter-Sectoral Impact Model Intercomparison Project (ISIMIP). The within-country EDC distribution was assumed to follow that of ethene. The reaction of ethene with chlorine is the main route by which industrial EDC production occurs and thus ethene should be a reasonable proxy. Anthropogenic ethene emissions (year 2014) from the 'industrial combustion and processes' sector were taken from the gridded (0.5°×0.5°) datasets produced for CMIP6 (Feng et al., 2020). **Figure 2** illustrates the resulting surface EDC emission distribution and timeseries of regional and global emissions for 'scenario sc05', i.e. with $\alpha_1 = 0.5\%$ (non-A5) / 1.5% (A5). In this example case, Asia (sum of Regions 3-5) accounts for ~48% of global emissions in 2020. For other Cl-VSLS, Asian emissions have been assessed to dominate the global anthropogenic source, such as the estimated ~90% contribution of Asia to global $CH_2Cl_2$ emissions reported by Claxton et al. (2020). For EDC, the approach and information described above gives rise to a more even distribution of emissions between continents, including a sizeable source outside of Asia.

**Table 3.** *Estimated global EDC emissions (Gg) due to fugitive losses between 2002 and 2020 (assuming 100% of EDC consumption is for feedstock use) and for different assumed production emission factors ($\alpha_1$) in non-Article 5 (developed) and Article 5 (developing) countries.*

| Scenario | $\alpha_1$ (%) | | Global EDC emission (Gg / yr) | | | | | | | | | |
|---|---|---|---|---|---|---|---|---|---|---|---|---|
| | *non-A5* | *A5* | *2002* | *2004* | *2006* | *2008* | *2010* | *2012* | *2014* | *2016* | *2018* | *2020* |
| **sc01** | **0.1** | **0.3** | 105 | 111 | 116 | 114 | 120 | 123 | 127 | 136 | 150 | 146 |
| **sc02** | **0.2** | **0.6** | 166 | 175 | 185 | 183 | 193 | 199 | 204 | 218 | 241 | 236 |





| | | | | | | | | | | | |
|---|---|---|---|---|---|---|---|---|---|---|---|
| **sc03** | **0.3** | **0.9** | 227 | 239 | 254 | 251 | 266 | 274 | 282 | 301 | 333 | 325 |
| **sc04** | **0.4** | **1.2** | 288 | 304 | 323 | 320 | 339 | 349 | 359 | 384 | 425 | 415 |
| **sc05** | **0.5** | **1.5** | 349 | 368 | 392 | 389 | 412 | 425 | 437 | 467 | 516 | 505 |
| **sc06** | **0.6** | **1.8** | 410 | 433 | 461 | 458 | 485 | 500 | 515 | 549 | 608 | 594 |

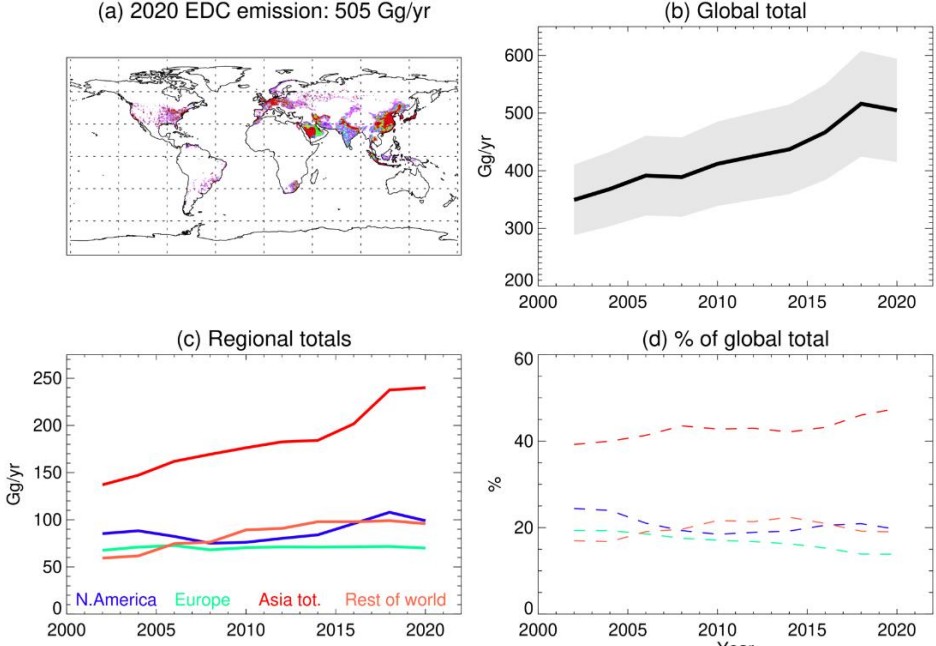

*Figure 2.* *(a) Estimated global surface EDC emission distribution in Jan 2020 ($10^{-2}$ kg/m²/s). (b) Global total EDC emission vs year (Gg/yr). (c) Regional total EDC emissions vs year. (d) Proportion of global EDC emissions by region (%). All data based on scenario sc05 (i.e. assuming $\alpha_1 = 0.5$ or 1.5%) with the lower/upper uncertainty (grey shading) in panel b denoting*
*the sc04 and sc06 cases.*

**2.3 CTM and experiments**

The time-varying EDC emissions described in Section 2.2 were included in the TOMCAT/SLIMCAT 3-D CTM (Chipperfield, 2006). The model is well evaluated and has been widely used to study the atmospheric budget and impacts of
a range of trace gases, including VSLS (e.g. Claxton et al., 2020; Hossaini et al., 2019). The offline model (hereafter 'TOMCAT') is forced by meteorological fields from the European Centre for Medium-Range Weather Forecasts (ECMWF) ERA5 dataset (Hersbach et al., 2020). The model uses the Prather (1986) scheme for tracer advection and the Holtslag and Boville (1993) scheme to represent boundary layer mixing. For convective transport, the model here utilised archived



convective mass fluxes from ERA5. This approach was previously evaluated for ERA-Interim by Feng et al. (2011) and found to perform well. All simulations were performed at a 2.8°×2.8° (T42 Gaussian grid) horizontal resolution and with 60 hybrid sigma-pressure (σ-p) levels extending from the surface to ~60 km.

To test model-measurement agreement under different EDC scenarios, and thus provide constraint on the global EDC source strength, a reduced chemistry configuration of the CTM was used. In this configuration, the concentration of the hydroxyl radical (OH) was prescribed from the monthly-varying climatology produced for the Tracer Transport Model Intercomparison Project (TransCom) on methane (Patra et al., 2011). Six simulations were performed in which the model EDC tracer was controlled by differing surface emissions (**Table 3**), along with transport, and oxidation by OH. The rate constant for the EDC + OH reaction ($k_{OH}$) was specified from the latest Jet Propulsion Laboratory (JPL) kinetics evaluation (Burkholder et al., 2019): $k_{OH} = 1.14×10^{-11} \exp(-1150/T)$. The reaction was assumed to proceed as: EDC + OH → 2Cl + products. There is no current recommendation given for EDC absorption cross sections and, like other chlorinated VSLS, photolysis is expected to be a minor sink and so was not considered.

For diagnosing EDC's contribution to stratospheric chlorine and its effect on ozone, we used a CTM configuration with 'full' stratospheric chemistry. This configuration includes a treatment of all major processes that control polar and extra-polar ozone (e.g. Chipperfield et al., 2018). Full chemistry simulations considered only the most likely range of EDC emissions (determined in Section 3.1) and the EDC tracer was treated as described above. In the troposphere, chlorine atoms released from EDC oxidation will quickly speciate into HCl, the dominant inorganic chlorine ($Cl_y$) reservoir (e.g. via $CH_4$ + Cl → HCl + products). Tropospheric removal of HCl and other $Cl_y$ species (HCl, HOCl and $ClONO_2$) through wet and dry deposition was calculated with the standard tropospheric TOMCAT routines (Giannakopoulos et al., 1999; Monks et al., 2017). Henry's law data used to calculate wet deposition rates were taken from Sander et al. (2023). Full chemistry model runs were spun-up for 10 years and then run over the full 19-year analysis period (2002-2020). The stratospheric chlorine and ozone response to EDC emissions were diagnosed from paired simulations (i.e. comparing runs with EDC emissions to a no-EDC control run). Time-varying surface mixing ratios of long-lived gases (halogenated ODSs, $N_2O$, $CH_4$ etc) were prescribed from the data given in WMO (2018).

**2.4 EDC observations**

We have utilised a range of aircraft measurements of EDC to evaluate the model and to provide constraints on the global EDC source strength. The HIAPER Pole-to-Pole Observation (HIPPO) mission (e.g. Wofsy et al., 2011) was conducted between 2009 and 2011 and involved measurements of a wide range of trace gases predominately over the Pacific from on board the National Science Foundation (NSF) Gulfstream V aircraft. Sampling extended over a large latitude range from roughly the North Pole to the Antarctic Ocean and from the surface to ~14 km. The mission comprised 5 campaigns conducted in different seasons: HIPPO-1 (January 2009), HIPPO-2 (November 2009), HIPPO-3 (March-April 2010),



HIPPO-4 (June 2011) and HIPPO-5 (August/September 2011). Measurements of EDC were obtained by the University of Miami based on the analysis of whole air samples collected in flask samples during each campaign.

The more recent NASA Atmospheric Tomography (ATom) mission was conducted between 2016 and 2018 also involved extensive measurements of trace gases from near pole to pole, including over the Pacific and the Atlantic. Measurements were obtained up to an altitude of ~12 km on board the NASA DC-8 aircraft over 4 campaigns covering different seasons:

ATom-1 (July-August 2016), ATom-2 (January-February 2017), ATom-3 (September-October 2017), and ATom-4 (April-May 2018). An overview of the ATom mission including some scientific highlights is given in Thompson et al. (2022). For this study, we have used EDC measurements obtained by NOAA from air samples collected with the Programmable Flask Package (PFP) whole air sampler. The EDC measurement precision is around 1% on average (for mole fractions of 1-2 ppt) and the detection limit is <1 ppt. Like HIPPO, the spatial coverage of ATom makes it an especially useful dataset with which

to evaluate global models (e.g. the representation of hemispheric gradients). In our subsequent analysis, data from both HIPPO and ATom have been aggregated into 9 latitude bins (>80°N, 60-80°N, 40-60°N, 20-40°N, 0-20°N, 0-20°S, 20-40°S, 40-60°S and <60°S). The mean, standard deviation, and number of datapoints for each bin (boundary layer only, <3km) is given in **Table 4**.

To examine model performance over Asia, we also used measurements obtained during the 2016 Korea-United States Air

Quality Study (KORUS-AQ) mission. The mission took place in the months of May and June and included 20 research flights of the NASA DC-8 aircraft based from Osan Air Base, approximately 50 km south of Seoul, South Korea (Crawford et al., 2020). Measurements during this campaign targeted local urban sources of photochemical pollutants and therefore the air sampled differs considerably from that sampled during HIPPO and ATom. Measurements of EDC and other gases were obtained from whole air samples collected by the University of California, Irvine (UCI) from the surface up to an altitude of

~11 km. The measurement detection limit was 0.1 pptv and the measurement precision was 5% (Simpson et al., 2020). Highly elevated levels of EDC and other VOCs were reported from KORUS-AQ, especially in airmasses originating from China (Simpson et al., 2020). For the analysis here, the EDC mole fractions were aggregated into 8 altitude bins (0-8 km) of 1 km depth. Sampling was extensive and the number of measurements in each bin ranged from 94 to 1323.

To further evaluate model performance over eastern Asia, we used surface EDC measurements made by the University of

East Anglia (UEA) at Bachok Marine Research Station, which lies on the North East of Malaysia (6.009°N, 102.425°E), and at two sites in Taiwan: (1) Fuguei Cape, on the northern Taiwanese coast (25.297°N, 121.538°E), and (2) Hengchun, on the southern coast (22.0547°N, 120.6995°E). Measurements of EDC and other Cl-VSLS at each location have been reported by Oram et al. (2017) and show elevated levels with respect to data obtained in other world regions. The same study provides full details of the sampling and instrument method. Briefly, measurements were obtained by whole air samples collected

between 2014 and 2020. Sampling is seasonal and targeted primarily at observing emissions from East Asia during the NE monsoon. Sampling at Bachok occurred in the months of November to March, while sampling at Fuguei Cape (2014 and 2016 onwards) and Hengchun (2013 and 2015 only) mostly occurred in March to May (**Table S1** in Supplement). The latter





two sites in Taiwan are combined to give a single time-series in our subsequent analysis. All collected samples were analysed by gas chromatography-mass spectrometry (GC-MS) at UEA, with a typical precision of 1-3 %.


*Table 4. Observed (Obs.) and modelled (Model) mean EDC abundance (ppt) in the boundary layer (< 3 km) and in different latitude bins during HIPPO and ATom. n denotes the number of measurements in each bin. Mean EDC is reported with ±1 s.d. The mean bias (MB, model minus observation) is given for each bin. Model results are based on EDC emission scenario sc05.*

| Lat. Bin | HIPPO campaign | | | | ATom campaign | | | |
|---|---|---|---|---|---|---|---|---|
| | *n* | *Obs.* | *Model* | *MB* | *n* | *Obs.* | *Model* | *MB* |
| **>80°N** | 11 | 13.3 (±4.5) | 12.6 (±4.7) | -0.7 | 2 | 17.7 (±2.6) | 20.3 (±3.0) | 2.6 |
| **60-80°N** | 77 | 14.8 (±4.5) | 12.5 (±4.5) | -2.3 | 40 | 16.4 (±3.5) | 19.3 (±4.0) | 2.9 |
| **40-60°N** | 67 | 15.2 (±4.2) | 13.0 (±4.7) | -2.2 | 55 | 17.3 (±5.9) | 18.6 (±4.4) | 1.3 |
| **20-40°N** | 53 | 15.9 (±6.7) | 11.1 (±4.8) | 4.8 | 28 | 16.4 (±8.3) | 17.0 (±6.4) | 0.7 |
| **0-20°N** | 49 | 8.5 (±3.7) | 6.0 (±3.1) | -2.5 | 42 | 8.9 (±5.1) | 7.5 (±4.2) | -1.3 |
| **0-20°S** | 40 | 3.5 (±1.2) | 2.3 (±0.9) | -1.2 | 25 | 3.5 (±1.2) | 2.7 (±0.7) | -0.7 |
| **20-40°S** | 68 | 1.9 (±0.5) | 1.9 (±0.6) | 0.02 | 21 | 1.9 (±0.3) | 2.2 (±1.1) | 0.3 |
| **40-60°S** | 36 | 1.9 (±0.4) | 2.1 (±0.6) | 0.2 | 35 | 1.8 (±0.3) | 2.1 (±0.6) | 0.4 |
| **<60°S** | 12 | 1.8 (±0.4) | 2.0 (±0.6) | 0.2 | 22 | 1.7 (±0.2) | 2.1 (±0.5) | 0.4 |

**3. Results and Discussion**

**3.1 Model-measurement comparison and emission constraint**

Observed boundary layer EDC mole fractions (<3 km), as averages from all deployments of HIPPO (2009-2011) and ATom (2016-2018), are shown in **Figure 3a** and **b**. The measurement data were compiled into 9 latitude bins (Section 2.4) and the mean (±1s.d.) of each bin is shown. Recall that both missions sampled air in various seasons of the year and thus the s.d.
variability includes seasonal effects. Measurements from both missions show a strong hemispheric gradient, with mean NH mole fractions of ~15 ppt at latitudes greater than 40°N and ~4 ppt or less in the Southern Hemisphere (SH). For comparison, EDC mole fractions of 7.8 (±1.5) ppt have been previously reported in background air in the NH based on aircraft measurements from the 2006 NASA INTEX-B mission which sampled around the Gulf of Mexico and over the West Pacific off the US coast (Barletta et al., 2009; Singh et al., 2009). The data in **Figure 3** show that variability is large within the NH,
where most EDC production (and emission) is located and where the number of measurements is relatively large. For




example, the relative standard deviation of the 20°-40°N bin is ~40% (HIPPO) and ~50% (ATom). For comparison, the relative standard deviation in the SH is smaller; e.g. ~21% (HIPPO) and ~12% (ATom) for the <60°S bin.

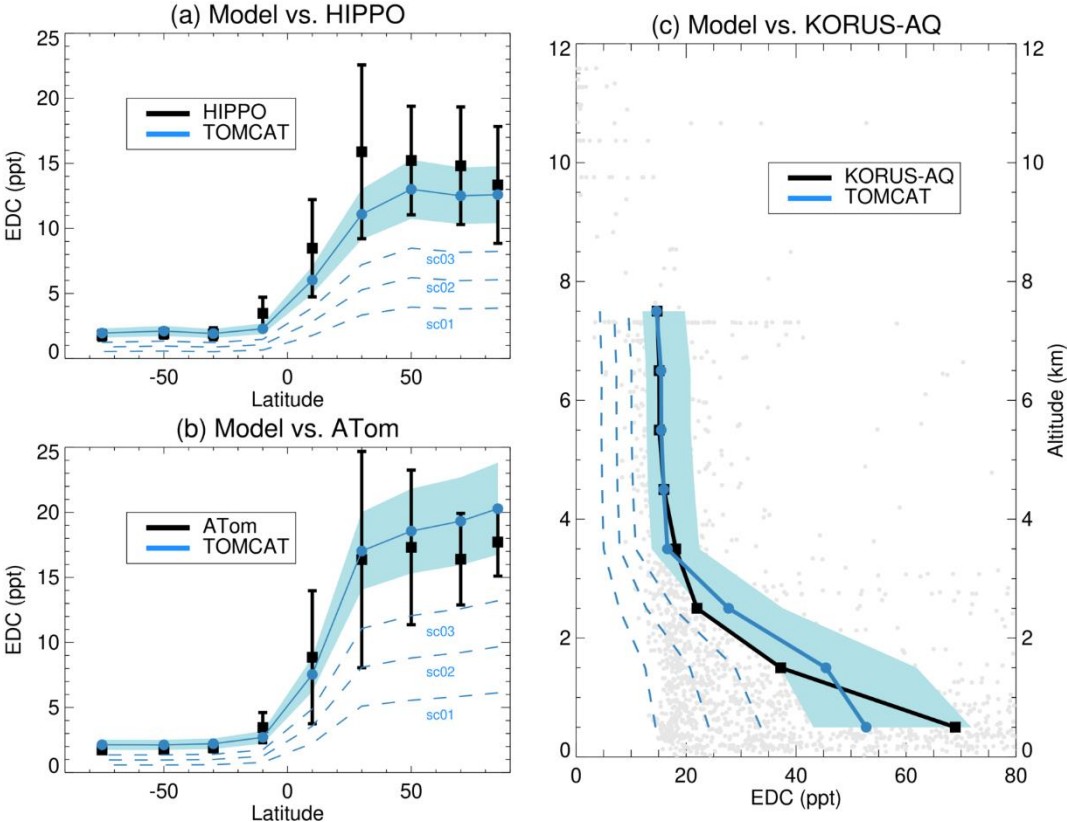


***Figure 3.*** *(a-b) Observed EDC mole fractions (ppt) as a function of latitude averaged over all deployments of each HIPPO (2009-2011) and ATom campaign (2016-2018). The data were obtained at < 3 km altitude and have been averaged in 9 latitude bins (see Section 2.4). Filled black symbols represent the mean within each bin (plotted at the central latitude) and error bars denote ±1s.d. The corresponding modelled EDC abundance from TOMCAT is shown for different assumed*

*emission factors. Dashed lines denote scenarios sc01 through to sc03. The blue shaded region denotes the range obtained from sc04 to sc06 with the central sc05 case indicated (solid line). (c) Observed EDC mole fractions vs altitude from the 2016 KORUS-AQ mission. All data from all flights (filled grey circles) were aggregated into 8 altitude bins (see Section 2.4) with the median of each bin shown (solid black line). Data points extend to 2.5 ppb but have been cut off at 80 ppt. The corresponding EDC abundance from TOMCAT (also median) is shown as for other panels.*



**Figure 3** also includes the TOMCAT modelled EDC abundance using the different emission scenarios, i.e. assuming different $\alpha_1$ emission factors (see Section 2.2 and **Table 3**). With scenarios sc01 through to sc03 the model exhibits a substantial low bias and is unable to reproduce the magnitude of EDC in either hemisphere or the observed hemispheric gradient from each mission. Better model-measurement agreement for both missions is obtained from runs with scenarios sc04 through to sc06, shown by the shaded regions in **Figure 3**. The statistics describing model-measurement differences in **Table 4** are based on the central sc05 case, i.e. $\alpha_1 = 0.5\%$ (non-A5 countries) / 1.5% (A5). Under this scenario, the mean bias (model minus observation) varies by latitude and ranges from near zero up to 4.8 ppt. Although underestimating mean EDC observed in the NH during HIPPO, the model falls within the measurement variability, and better agreement is obtained for the comparisons with ATom. Generally, model-measurement biases here are difficult to interpret and could in part reflect the model OH field (affecting the EDC lifetime) and/or transport processes; they do not necessarily point to an under or overestimation of local emissions. Additionally, as for some other VSLS, differences in calibration scales between measurement groups could be a confounding factor (see Roozitalab et al., 2024 for a more detailed discussed). Importantly, model-measurement agreement is generally good in the tropics ($\pm20°$ N/S), the region most relevant for diagnosing transport to the stratosphere, throughout the vertical profile (see Supplement **Figure S1**). The low abundance of EDC at SH high latitudes is also well captured.

A comparison of the modelled vertical profile of EDC to that observed during KORUS-AQ (2016) is shown in **Figure 3c**. Compared to HIPPO and ATom, far larger observed levels of EDC are apparent at lower altitudes (up to 2.5 ppbv; not shown in Figure 3c), along with very large variability (see filled grey circles). To accommodate the latter, binned measurement data in **Figure 3c** show the median as opposed to the mean and the horizontal axis is capped at 80 ppt. The maximum observed value of >2.5 ppb occurred in the 0-1 km bin for air originating from China. A similar maximum of >2.4 ppb was measured in air originating from an industrial facility in South Korea (Simpson et al., 2020), though we note that a global scale model is not expected to capture these most extreme values, which included targeted source sampling. Although the model (median ~53 ppt under scenario sc05) underestimates the observations (median ~69 ppt) in the lowest bin (0-1 km), it is evident from comparing **Figure 3c** with **3a** and **3b** that the model shows significantly elevated EDC in this region. Above 1 km, there is very close agreement between the model and measurements using scenario sc05. A full quantitative comparison is given in **Table S2** in the Supplement. A previous in-depth analysis of the KORUS-AQ data highlighted that EDC was especially elevated in air originating from China (Simpson et al., 2020). Similarly, during the 2006 NASA INTEX-B mission, analysis of air sampled in polluted plumes from Asia (though especially China) revealed substantially elevated EDC relative to background air and to plumes from the USA (Barletta et al., 2009). Indeed, EDC was used as a tracer of air from China during both INTEX-B and KORUS-AQ.

The modelled EDC abundance is compared to the available surface measurements from Bachok and Taiwan in **Figure 4**. As above, the measurements are characterised by large variability with EDC exceeding 150 ppt at Bachok and 300 ppt at Taiwan on some days. While the model is not expected to capture the most extreme values, the central tendency of the



observations appears to be reasonably well captured at both sites under scenario sc05 (see **Table S1** in Supplement). Both the measurements and model show especially elevated levels of EDC at Taiwan (median values >50 ppt in each year of
sampling) with respect to levels observed during HIPPO and ATom, suggesting strong regional or local sources and sampling of relatively polluted air in both the Bachok and Taiwanese samples. Samples collected at Bachok, where the model captures the shape of the seasonal cycle well, predominately occur when the site experiences north-easterly winds and observations are thus likely impacted by emissions occurring from mainland China (Oram et al. 2017). Similarly, owing to the close proximity, measurements at Taiwan are expected to be influenced by emissions from mainland China. Although
not exhaustive, these comparisons (along with those for KORUS-AQ above) suggest the model has a reasonable representation of regional emissions in East and South East Asia. Note, at Taiwan (only) the modelled EDC abundance was found to exhibit a strong sensitivity to the choice of model vertical level sampled. Model data in **Figure 4b** therefore represent the average of the two model levels closest to the surface.

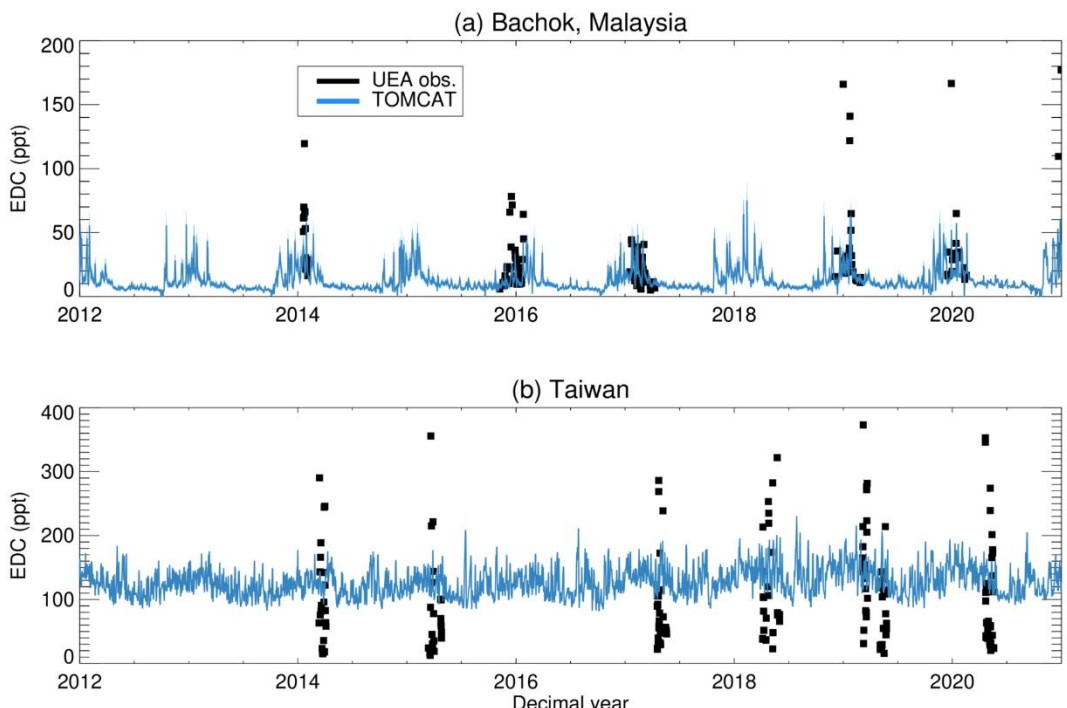


***Figure 4.** Observed EDC surface mole fraction (ppt) at (a) Bachok and (b) Taiwan. The corresponding modelled EDC abundance from TOMCAT (sampled daily) is shown for scenario sc05, with blue shaded regions denoting the range from scenarios sc04 and sc06.*






Although the regional variability of atmospheric EDC measurements can be large, the remote atmospheric survey sampling represented by the HIPPO and ATom missions, along with the other comparisons, allows for some constraint on the global EDC source (given our assumptions concerning the EDC emission distribution). Assuming scenario sc05, $\alpha_1 = 0.5\%$ (non-A5 countries) / 1.5 % (A5 countries), shown above to provide reasonable model-measurement agreement, we estimate a
global EDC source of 349 (±61) Gg/yr in 2002, rising to 505 (±90) Gg/yr in 2020 (i.e. an increase of ~45%). There are very few estimates of global or regional EDC emissions in the literature with which to compare these findings. Using a simple tracer ratio method, Wang et al. (2014) estimated Chinese EDC emissions of 121.6 (±89) Gg/yr in 2010. For the same year and again utilising scenario sc05 (range sc04 to sc06), our inventory produces significantly lower Chinese EDC emissions of ~44 (36-52) Gg/yr. Oram et al. (2017) estimated Chinese EDC emissions of 203 (±9) Gg/yr for 2015 based on measurements
obtained in Taiwan and Malaysia. Our estimate for Chinese emissions in 2016 is ~60 (49-71) Gg/yr and is thus substantially lower. However, it should be emphasised that inferred emissions from observed tracer correlations, as in the above studies, are based on several assumptions and subject to large uncertainty. For instance, emissions occurring in nearby regions may confound geographical attribution. Although still lower, our inventory provides better agreement to the above estimates if emissions from, for instance, nearby Taiwan are included. We estimate the sum of EDC emissions from China and Taiwan to
be 89 (73-106) Gg in 2010 and 107 (87-127) Gg in 2016. A summary of these comparisons of Chinese emissions is given in **Table S3** of the Supplement. A further factor that may confound the comparison of these different estimates is the month of measurement and sampling frequency used to infer tracer ratios. For example, note that the EDC measurements at Bachok reported by Oram et al. (2017), an extended timeseries of which are shown in **Figure 4a**, focus on non-summer months when EDC is relatively abundant at that site.

While the above results provide some constraint on the global EDC source required to reproduce atmospheric observations, some care is needed when interpreting the findings from an emission process standpoint. Our analysis has approximated fugitive emissions arising from production, feedstock use and distribution, assuming feedstock uses account for all consumption in every country. However, EDC has known solvent uses (not explicitly accounted for) which may be up to 100% emissive in the absence of solvent capture and careful disposal. If a non-negligible amount of the observed
atmospheric abundance of EDC stems from solvent use, then the contribution from fugitive losses could be overestimated. We anticipate that EDC solvent use is most prevalent in developing countries where it is relatively cheap versus alternatives and is readily available, and where concerns over its toxicity may not yet have resulted in restrictions on its use. Given these uncertainties, we do not overinterpret our findings from an emission process or sectoral standpoint but rather, with more confidence, highlight the overall magnitude of emissions that provide good agreement with the available measurement data
in **Figure 3**. In subsequent sections, we present all model quantities assuming scenario sc05 emissions, with reported uncertainties from the sc04 and sc06 cases.



## 3.2 Lifetime, tropospheric distribution, and contribution to stratospheric chlorine

The modelled tropospheric distribution of EDC is shown in **Figure 5** for the years 2002 and 2020. At the surface (panels a and b) EDC exhibits large spatial variability and has a strong hemispheric gradient. Hotspots occur within the industrialised zones of its main source regions (USA, Europe, East Asia). Recall, the EDC emission distribution within countries is prescribed here to follow that of ethene. In reality, the EDC source may be less dispersed than assumed, particularly if fugitive emissions occur from a relatively small number of point plant locations. Growth in the NH background of EDC is apparent from **Figure 5** and we estimate that the global EDC burden increased from ~81 (±15) Gg to ~116 (±21) Gg between 2002 and 2020 (**Table 5**). Chemical loss of EDC is controlled primarily through reaction with OH and we calculate an overall global EDC lifetime, defined as the ratio of its global burden over its global loss rate, to be ~83 days in 2020 (**Table 5**). This is in very close agreement to the ~82 days reported by Burkholder and Hodnebrog (2022).

It is well established that VSLS may contribute to stratospheric halogen loading via both source gas injection (SGI) and product gas injection (PGI). Chlorine SGI and PGI from EDC are also shown in **Figure 5**. Defining these quantities at the tropical tropopause (~17 km), the total (SGI+PGI) stratospheric chlorine input from EDC in the year 2020 is estimated to be 12.9 (±2.4) ppt Cl, comprising 10.7 (±2) ppt Cl from SGI and 2.2 (±0.4) ppt Cl from PGI (**Table 5**). For context, the total Cl-VSLS supply to the stratosphere (including VSLS other than EDC) was estimated to be ~130 (100-160) ppt Cl in 2019 when total stratospheric chlorine (i.e. including long-lived gases) was around 3240 ppt Cl (Laube and Tegtmeier et al., 2020). Our EDC SGI estimate in 2020 is similar to the 8.5 (±1.9) ppt Cl reported in our previous modelling work (that did not include geographically- or time-varying emissions) for the year 2017 (Hossaini et al., 2019). A notable difference with our previous work is that here we assess that stratospheric chlorine from EDC has increased significantly over time (see **Figure S2**), reflecting growth in emissions and hence SGI. In the current study, our estimated chlorine PGI is also similar to the value of ~2 ppt Cl reported in Hossaini et al. (2019). The latter assumed a fixed lifetime of $Cl_y$ in the troposphere against deposition (~5 days), while here we adopted an improved, more explicit representation in which $Cl_y$ washout was calculated using the standard TOMCAT deposition routines for the component chlorine species (Section 2.3). As for all VSLS, lack of observational constraint means that modelled PGI estimates carry significant uncertainty. A process not considered here is the heterogeneous recycling of $Cl_y$ on ice crystals in the upper troposphere, for which there is some observational evidence (von Hobe et al., 2011). As demonstrated for iodine (Saiz-Lopez et al., 2015), such a process could plausibly extend the $Cl_y$ lifetime and thus increase the magnitude of PGI. However, the processes and the required parameters with which to treat them in a global model are uncertain.

The model estimates of chlorine SGI from EDC that are presented in **Figure 5** (c and d) and **Table 5** are annual mean quantities at the tropical tropopause, averaged zonally over the whole of the tropics (±20°N/S). While transport across the tropical tropopause is the main route via which air enters the stratosphere, relatively elevated levels of VSLS (and other gases) have been reported in the subtropical NH lower stratosphere owing to the effects of the Asian summer monsoon (ASM) and ASM anticyclone (e.g. Fiehn et al., 2017; Keber et al., 2020; Lauther et al., 2022). Forming in boreal summer,



ASM dynamics are characterised by rapid uplift of boundary layer air to the UTLS by deep convection (e.g. Randel and
Park, 2006; Basha et al., 2020), including relatively polluted air masses from South and East Asia (e.g. Li et al., 2005;
Randel et al., 2010; Müller et al. 2016). Based on aircraft measurements obtained during the Asian Monsoon Anticyclone
2017 campaign (AMA-17) over the Indian subcontinent, Adcock et al. (2021) reported a mean EDC mole fraction around the
tropopause (355-375 K) of ~12 ppt (with a range of 4.5 to 23 ppt), corresponding to a chlorine SGI (i.e. 2× the EDC mole

fraction) in the range 9-47 ppt Cl. The AMA-17 measurements were obtained in July and August in the latitude range 21°N-
29°N, longitude range 79°E-91°E, and from ~10-20 km altitude. The measurements from Adcock et al. (2021) are shown in
**Figure S3** of the Supplement along with corresponding model estimates. There is generally good agreement between the two
datasets and the model corroborates the signal of relatively large levels of EDC around the tropopause (~10 ppt) and hence a
larger local stratospheric chlorine SGI (~20 ppt Cl) in this region/season relative to the annual mean quantities around the

450 tropical tropopause reported in **Table 5**.

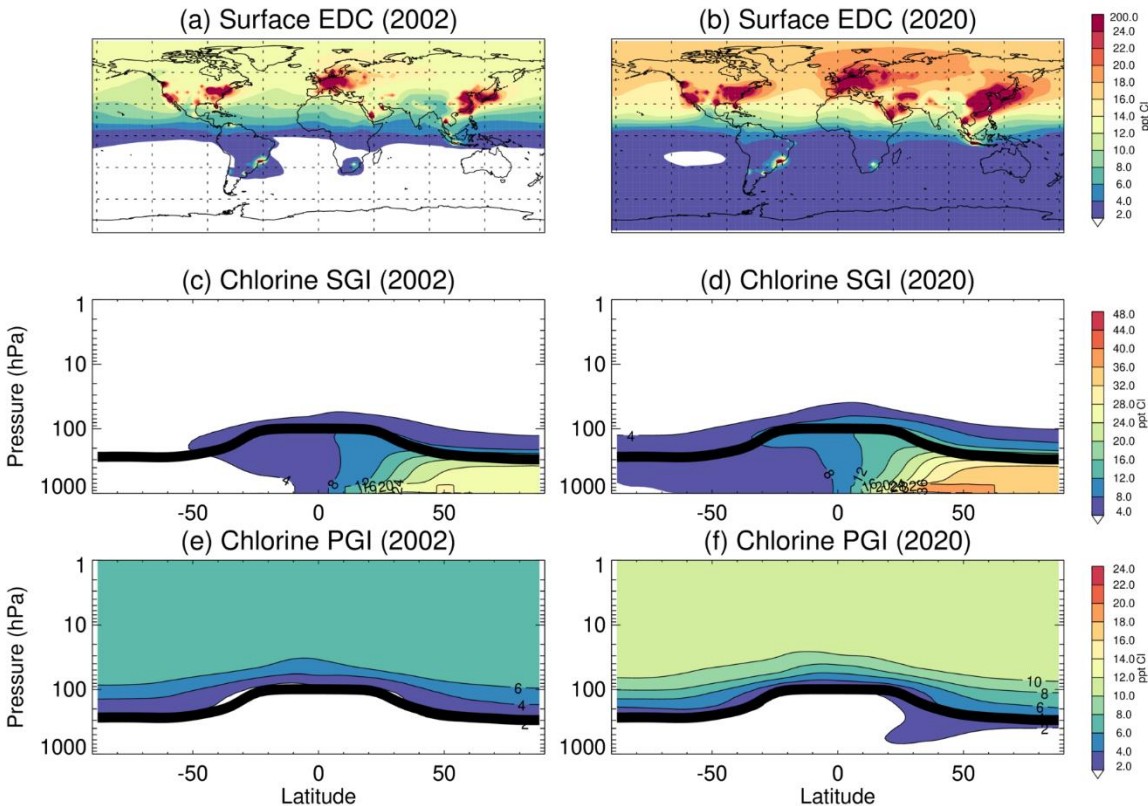

***Figure 5.*** *Modelled annual mean EDC volume mixing ratio (ppt) at the surface under scenario sc05 in (a) 2002 and (b)
2020. Panels (c-d) show the latitude-pressure distribution of chlorine SGI from EDC (ppt Cl) for the same years. Panels (e-
f) show chlorine PGI from EDC (ppt Cl). The thermal tropopause pressure based on ERA5 reanalysis (Hoffmann and Spang,*

*2022) is shown by the black line.*





*Table 5. Modelled EDC burden (total mass), global loss rate (due to OH), overall global lifetime (burden / loss rate) and contribution to stratospheric chlorine (ppt Cl) through SGI, PGI and total (SGI + PGI). All fields are annual averages for the years 2002 or 2020 and results are shown for emission scenarios (sc) sc04, sc05 and sc06.*

| Sc | Burden (Gg) | | Loss rate (Gg/yr) | | Lifetime (days) | | SGI (ppt Cl) | | PGI (ppt Cl) | | Total Cl (ppt Cl) | |
|---|---|---|---|---|---|---|---|---|---|---|---|---|
| | *2002* | *2020* | *2002* | *2020* | *2002* | *2020* | *2002* | *2020* | *2002* | *2020* | *2002* | *2020* |
| **04** | 67 | 95 | 289 | 417 | 85 | 83 | 5.5 | 8.8 | 1.2 | 1.8 | 6.7 | 10.6 |
| **05** | 81 | 116 | 350 | 508 | 85 | 83 | 6.7 | 10.7 | 1.5 | 2.2 | 8.2 | 12.9 |
| **06** | 96 | 137 | 411 | 598 | 85 | 83 | 7.9 | 12.7 | 1.7 | 2.6 | 9.6 | 15.3 |

## 3.3 Impact of EDC emissions on ozone

The modelled stratospheric ozone change due to EDC under 2020 conditions is shown in **Figure 6**. EDC decreases stratospheric ozone globally, though the effect is generally small. The largest absolute decreases occur in the upper stratosphere (10-1 hPa) and polar lower stratosphere (200-20 hPa), i.e. regions where chlorine-catalysed ozone loss is known to be important (e.g. Chipperfield et al., 2018). The absolute ozone decreases in **Figure 6** (panels a-b) are up to ~5 ppb when expressed as an annual average (panel a). Larger decreases (up to ~10 ppb) occur within SH high latitudes in Spring when the Antarctic ozone hole forms (panel b). Corresponding ozone changes expressed in percent are shown in panels c-d. In most regions, the ozone changes due to EDC represent changes of <1%, though in the SH polar spring reductions of up to ~1.3% in the lower stratosphere are found.

The small (though non-zero) effect of EDC on global stratospheric ozone reflects the relatively small input of chlorine from EDC to the stratosphere (see above). However, as noted above, several studies have identified transport via the ASM as a route through which relatively large local injections of various VSLS (including EDC) to the extratropical lower stratosphere can occur (Keber et al., 2020; Adcock et al., 2021; Lauther et al., 2022). In principle, this process and its effect on chlorine injection from EDC is represented in our model (see **Figure S3**). However, the very large surface levels of EDC, in at least some parts of Asia (see **Figure 2c**), that are not tightly constrained by the data considered here may be underestimated in this analysis and thus too the co-location of emission hotspots with regions of relatively fast vertical ascent. The significance of the Asian Summer Monsoon transport pathway for VSLS-driven stratospheric ozone loss is an area of current research and will require further and more detailed investigation once new measurements in this region become available.

This study has focussed only on the possible direct impact of EDC emissions on ozone. However, we note that a broader impact assessment (beyond the scope of this work) might also factor in the unintended, but expected to be very minor, formation of other halogenated chemicals that inevitably occur during the EDC production process (the majority of such are





destroyed by thermal oxidation or other means). These species are found in the "lights" and "heavies" effluent streams which may range from 0.3-1.0% of the EDC produced (TEAP, 2022), and which include a proportion of ODSs, such as carbon tetrachloride ($CCl_4$), and other chlorinated VSLS, such as chloroform ($CHCl_3$).

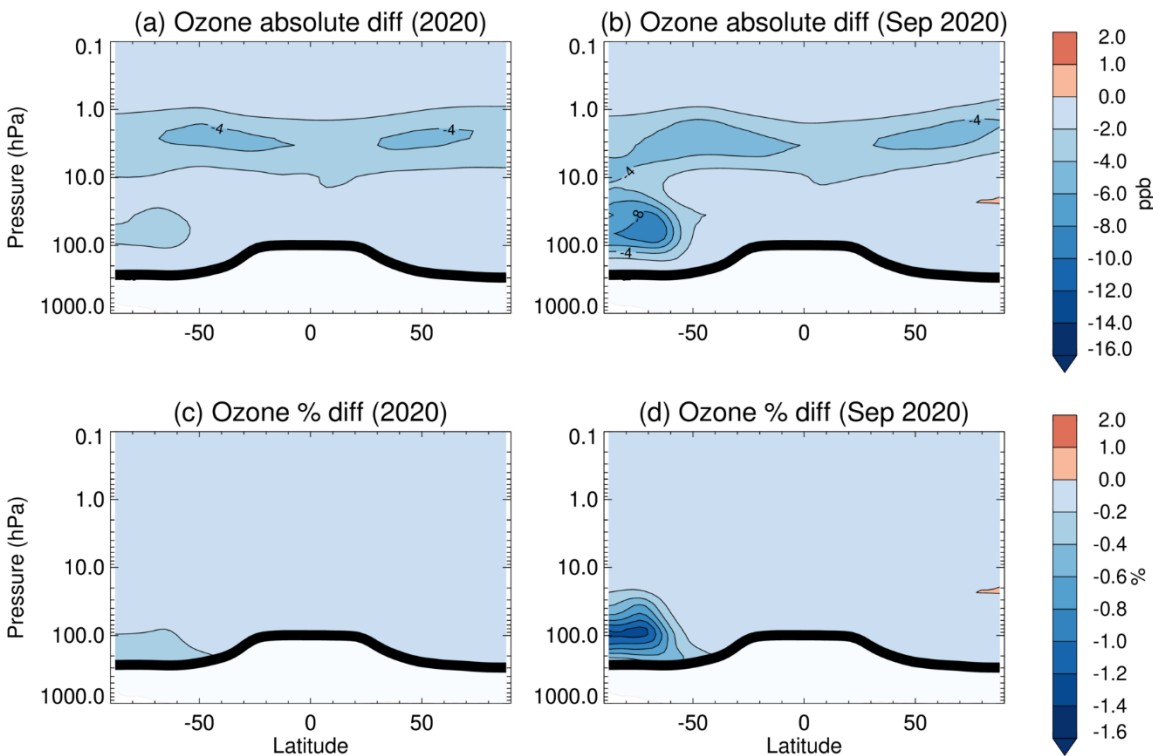

**Figure 6.** *Modelled stratospheric ozone decrease due to EDC in 2020 expressed as (a,c) an annual average, and (b,d) September average (i.e. Antarctic ozone hole season). Panels (a-b) show absolute decreases (ppb) and panels (c-d) percentages. Model results calculated based on the difference between simulations with EDC (scenario sc05) and without EDC.*

## 4. Summary and concluding remarks

The global production of EDC in the year 2020 exceeded 50 million tons. However, despite annual production volume greatly exceeding that of other more prominent industrial VSLS (e.g. $CH_2Cl_2$, $CHCl_3$), few atmospheric observations of EDC exist and little is known of its global budget. In this study, we combined information on industrial EDC production, trade statistics, and assumptions on its fugitive losses, to explore the plausible range of global EDC emissions. Time-varying gridded EDC emission fields were developed and then included in the TOMCAT CTM. Transient simulations were





performed to assess the EDC source required to reproduce a variety of measurements from recent aircraft missions (HIPPO, ATom, and KORUS-AQ) and from ground sites in South East Asia. Based on constraints provided by these comparisons, we infer a global EDC source of 349 (±61) Gg/yr in 2002, rising to 505 (±90) Gg/yr in 2020 (i.e. an increase of ~45%). Our framework for calculating EDC emissions assumed that all releases to the atmosphere result from fugitive losses during its production, its use as a feedstock (largely to produce VCM in the PVC production chain), and during its supply chain.

Reasonably good agreement between the model and EDC observations is achieved assuming a production emission factor of ~0.5% in developed countries and 1.5% in developing countries. These factors are within the generic 'most likely' range of factors (0.9-4.0%) applicable to a range of other gases assessed by TEAP (2022). Large uncertainty around the magnitude and emissions associated with EDC solvent use, which is potentially widespread in developing countries and East Asia, is a confounding factor in our analysis and prevents firm conclusions as to the specific sectors contributing to the observed EDC

signal and the global distribution of these emissions.

We estimate that EDC contributed 12.9 (±2.4) ppt of chlorine to the stratosphere in 2020. Based on this loading, we estimate EDC decreased ozone by up to several ppb in 2020, with the largest changes occurring in the upper stratosphere and high-latitude lower stratosphere. Outside of the SH lower stratosphere in Spring, where the ozone decreases attributable to EDC are up to ~1.3%, the effect of EDC on global stratospheric ozone is presently small (<1%), though non-zero. Any future

growth in EDC emissions (e.g. tied to downstream demand for PVC) may increase the contribution of EDC to stratospheric chlorine and thereby increase its impact on ozone. Such possible future effects would need to be examined with knowledge of the global PVC market and its possible future trajectories and an assessment of use and emissions from the EDC solvent sector. Diagnosing future changes in the contribution of EDC to ozone-depleting chlorine in the stratosphere would also benefit from routine observations of EDC at sites across the globe.



**Data availability**

The TOMCAT model output and gridded emission data will be uploaded to the Zenodo open access repository
(https://zenodo.org/) if the manuscript is accepted for publication following peer-review.

**Author contribution**

RH conceived and led the study and developed the emission inventories in collaboration with, and based on data and analysis
provided by, DS. ZW performed the TOMCAT/SLIMCAT model simulations supervised by MPC and WF. AM and AL
contributed to the analysis of model output. DO, KA, CC, SAM, IJS and EA provided atmospheric measurements and
interpretation of these data. RH prepared the original draft of the paper. All authors contributed to reviewing and editing of
the manuscript.

**Competing interests**
The authors declare that they have no conflict of interest.

**Acknowledgements**

RH, MPC and WF were supported by the NERC projects LSO3 (NE/V011863/1) and InHALE (NE/X003582/1). UEA
would like to thank Lauren Gooch and Debbie Sanchez for past assistance with sample analysis and Ahmad Amin Abdullah
for sample collection at the Bachok Marine Research Station. The long-term sampling programmes in Taiwan and Malaysia
were established through the NERC International Opportunities fund (NE/J016012/1, NE/N006836/1) and subsequently
supported through the NERC SISLAC (NE/R001782/1) and LSO3 (NE/V011863/1) projects. K.E.A. was funded by the UK
Natural Environment Research Council through the EnvEast Doctoral Training Partnership (Grant NE/L002582/1). SAM
acknowledges the assistance of those facilitating measurements and calibration scales at NOAA including B. Hall, F. Moore,
K. McKain, and C. Siso. EA acknowledges technical assistance from X. Zhu and L. Pope and financial support from NASA
grant #80NSSC22K1284 and NSF AGS grants #0959853 and #1853948.

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
