# Peer review of "On the atmospheric budget of 1,2-dichloroethane and its impact on stratospheric chlorine and ozone (2002-2020)"

_EGUsphere, 2024_

## Referee Comment (RC2)

**Review of the manuscript "On the atmospheric budget of ethylene dichloride and its impact on stratospheric chlorine and ozone (2002-2020)" by Hossaini et al., 2024.**

The manuscript presents the development of a new bottom-up emission inventory for 1-2 dichloroethane (EDC) with enhanced spatial and temporal resolution compared with previous studies. An increasing trend in the global annual flux of EDC between 2002 and 2020 is determined based on regionally distributed production and consumption data. The new inventory is used in TOMCAT to estimate the Source Gas (SGI) and Product Gas (PGI) Injection of chlorine to the stratosphere, which results in a small (<1%) but not negligible impact on stratospheric ozone. The paper is very well organized, referenced and written, and certainly of interest for the community. Therefore, I suggest the work to be accepted with minor revisions. In the attached document, I provide a couple of general comments that might help to enhance the work visibility, and a list of minor and/or technical comments to be addressed.

**Main comments:**

- The authors mention several times that large regional and seasonal EDC enhancements are predicted with the new inventory, particularly for the Asian Summer Monsoon (ASM), and provide support by referring to the literature as well as by comparing with observations (e.g., in lines L32-37; L84-87; L266; L335-336; L439). In doing so, they should provide a stronger connection with the results published in Roozitalab et al., 2024 (cited in the manuscript) as well as to the recent ACCLIP paper from Pan et al., 2024 (https://www.pnas.org/doi/10.1073/pnas.2318716121). Most importantly, I think it would be a great idea to provide an estimate of the impact of enhanced EDC in the ASM over lower stratospheric ozone during the summer (see specific comment below).

- Given that the inventory considered bi-annual data and you performed a complete simulation for almost 20 years, it would be great to provide the mean rate of growth of EDC both for the surface emission as well as for the SGI and PGI. Those trends values (properly quantified) will be of interest for future reports on VSL influence on stratospheric ozone. In case the trends for the 2002-2020 period differ significantly to the trend during the last 4-5 years, an explicit statement and quantification could be provided.

**Minor Comments:**

L20: "transport of EDC (or its atmospheric oxidation products)". Is this "or" or "and"?

L74: "… at reportedly both urban and background sites …". Please revise text.

L78: Please indicate for which year were estimated the EDC emissions in the refernced study.

L133: "However, …". This however seems to indicate discrepancy with previous results, but they all point in the same direction.

L146: "Although the imbalance was small compared to the large production volumes of EDC, …". By how much? At least a percentage number should be provided.

L239: please provide a reference supporting the neglecting of EDC photolysis. Note that other studies cited in this work (Roozitalab et al., 2024) considered photolysis for EDC.

L343: I found reasonable to show the median instead of the mean for this case. Just by how much do the mean and the median differ?

L348: I completely agree with the statement and the link with Fig. 3, but I feel that this should also be linked to the spatially heterogeneous source strength shown in Fig. 2a.

L362: "Samples collected at Bachok, where the model captures the shape of the seasonal cycle well, …". I'm really surprised about the large seasonal cycle (both observed and modeled) at this site. Why is this? Is it because the emissions also show a large variability? Is it because OH changes, is it because meteorology? Is it due to influence from continental China? Please extend about this interesting topic !!!

L414: Given by your methods description, reaction with OH is the only chemical loss in your model. Or does EDC also suffer any type of washout / dry-deposition?

L432-434: In addition to the referred work for Iodine chemistry, a simplified representation of these ice-recycling reactions has also been performed for bromine and chlorine in Fernandez et al., 2014 (https://doi.org/10.5194/acp-14-13391-2014). The impact of these and other reactions on PGI and ozone loss for the case of bromine was addressed in Fernandez et al., 2021 (https://doi.org/10.1029/2020GL091125), though no estimations was performed for chlorine PGI.

L475-477: I completely agree with the statement, and suggest that in order to advance in that area, providing an estimation of how much larger is the absolute and/or percentage ozone decrease within the ASM region would be of interest here.

L490-493: You should explicitly mention in the conclusion that you used a bottom-up approach.

L493-494: "Time-varying gridded EDC emission fields were developed and then included in the TOMCAT CTM." It would be great if you can provide the emission inventory to the community to evaluate it in other models.

**Figures and Tables**

Table 4: I understand that in the last row it should say ">60°S" instead of "<" … as southern latitudes are not negative but "South".

Figure 3: would it be possible to show an error bar (spread of data) for the observed data in the vertical profile (panel c)?

Figure 4: I do not see the shading (sc04 and sc06) but only output for the sc05 results. I think it would be very useful to provide the range here.

Figure 6: The text mention several times the importance of the high EDC emissions over Asia and the rapid transport due to the ASM, but then the figure highlights the influence during the Antarctic Spring. Wouldn't it be nice to show also delta O3 values for the ASM region during July or August?

---

## Author Response (AR1)

**On the atmospheric budget of ethylene dichloride and its impact on stratospheric chlorine and ozone (2002–2020)**

We thank both reviewers for their positive reviews and helpful comments. Reviewer comments are repeated in *italics* below and our responses to each are given in **blue**.

**Responses to Anonymous Reviewer #1**

*The paper by Hossaini et al al presents a detailed study of 1,2, Dichloroethane, also often referred to as ethylene dichloride. It uses a chemical transport model, atmospheric observations and estimates of production and emission rates to derive an atmospheric budget. The paper is well written and the assumptions and the methods are well justified. I have a few issues, which are more of technical nature concerning the data used which I would like the authors to clarify. One further issue is that I do not like the use of the name ethylenedichloride, which is not a systematic name and also not a specific name. I suggest to use 1,2-dichloroethane instead, which is the correct name and also clearer as it is clear that it is the unsymmetrically substituted compound. Most importantly, I would like the authors to discuss little bit more about the data quality (sample stability and comparison of calibration scales). Apart from that I only have a few minor issues below. I recommend the paper to be accepted after minor revisions.*

We thank the Reviewer for this comment around naming. Although 'ethylene dichloride' (EDC) is the common trade name, we agree that 1,2-dichloroethane ('DCE' for shorthand) is the preferred IUPAC name. The latter is also how the molecule is referred to in WMO/UNEP Scientific Assessment of Ozone Depletion reports. Therefore, we have amended the manuscript throughout accordingly (including the title).

*l. 203: is ethene mainly of anthropogenic origin? or are there significant natural sources?*

Ethene has significant natural and anthropogenic sources. However, for the purpose of this work, we consider only the anthropogenic emission distribution (as a proxy for the DCE emission distribution).

*section 2.4.: please include some information on the calibration scales used for the measurements and the comparability of the different observations from three different groups. Are they all on the same scale? Also, has the stability of 1,2-dichloroethane in the samples been analysed?*

As a universally adopted international DCE calibration scale is not available at the time of writing, there are likely some differences between groups. The NOAA DCE measurements from ATom are based on the "NOAA-2021" scale. The UCI group use a scale provided by the University of Miami. Details of the UEA scale are given in Oram et al. (2017): DCE "was calibrated at UEA using the established static dilution technique recently described (Laube et al., 2012)". These details have been added to the revised manuscript in the relevant places of Section 2.4.

A detailed examination of differences between the measurement groups is beyond the scope of the paper and will be led by the measurement community in forthcoming work. However, note that scales among the labs considered in this study have historically not differed by more than 10-30% for gases like DCE. To more directly address the reviewer's query, we have conducted a preliminary comparison of background atmospheric DCE mole fractions obtained at two remote sites (Barrow and Samoa Observatories) where both the NOAA and UCI groups sample. This informal intercomparison reveals an average offset of up to ~30% at these two sites. It should be emphasised that this comparison is limited in scope and a more formal and extensive examination

will be required that is beyond the scope of this work. In the revised manuscript we have added the following text at the bottom of Section 2.4:

"Compared to other Cl-VSLS, scientific interest in DCE from an ozone depletion perspective is relatively new. As such, an international standard calibration scale has not yet been universally adopted across measuring groups. Historically, the scales among the labs considered in this study have not differed by more than 10-30% for gases similar to DCE. However, in the absence of any formal assessment of calibration scale differences, an informal intercomparison for DCE was performed for this work. Background atmospheric DCE mole fractions from two remote sites (Barrow and Samoa Observatories), where both the NOAA and UCI groups sample, were compared (2017 – 2023). This intercomparison revealed an average offset of up to ~30% (UCI relatively high / NOAA relatively low), i.e. at the upper end of the above range. While this comparison is limited in scope and will require further effort to refine (beyond the scope of this paper), this uncertainty is highlighted in the ensuing discussion".

In addition to the above text, we will acknowledge more explicitly that model-measurement differences from mission to mission (e.g. Figure 3, Figure S1) could reflect differences in calibration scales used by the labs supplying measurements and to some degree may confound the assessment of model performance and emissions. We do already make a point along these lines (see line 335 of original manuscript) but will strengthen/expand it by pointing to the above findings. It is important to emphasise that calibration scale differences don't affect our main conclusions around the existence of substantial global DCE emissions.

Regarding sample stability, the NOAA ATom sampling was conducted via pressurization into glass flasks, and there has been no indication of systematic growth or destruction of DCE in glass flasks over time within the measurement precision. Similarly, the UCI group has run extensive tests on the stability of compounds in their canisters in the time between sampling and analysis, and EDC is stable within the canisters. UEA samples are collected in silco-treated cylinders (stainless steel with inner surface coated with fused silica, Restek) and no issues with sample loss have been noticed.

These points are also now made in Section 2.4 of the revised manuscript.

*l. 270: does this mean that only the data below 3 km of HIPPO and ATOM were used? Why have the free tropospheric data not been used?*

The model-measurement comparisons in Table 4 and Figure 3 (a,b) focus on the boundary layer as we are interested in comparing in the region of the atmosphere where the influence of surface DCE emissions will be clearest. However, note that we do also compare model-measurement vertical profiles from HIPPO and ATom in Figure S1.

*l. 371, Figure 4: I could not find the blue shaded region to represent the model scenario ranges.*

The shading is given but we agree that it is difficult to see. This is due to the high frequency of measurements. We will make this clearer in the revised manuscript.

**Responses to Reviewer #2 (Dr Rafael Fernandez)**

*The manuscript presents the development of a new bottom-up emission inventory for 1-2 dichloroethane (EDC) with enhanced spatial and temporal resolution compared with previous studies. An increasing trend in the global annual flux of EDC between 2002 and 2020 is determined based on regionally distributed production and consumption data. The new inventory is used in TOMCAT to estimate the Source Gas (SGI) and Product Gas (PGI) Injection of chlorine to the stratosphere, which results in a small (<1%) but not negligible impact on stratospheric ozone. The paper is very well organized, referenced and written, and certainly of interest for the community. Therefore, I suggest the work to be accepted with minor revisions. In the attached document, I provide a couple of general comments that might help to enhance the work visibility, and a list of minor and/or technical comments to be addressed.*

We thank Dr Fernandez for their positive comments and helpful suggestions.

***Main comments:*** *The authors mention several times that large regional and seasonal EDC enhancements are predicted with the new inventory, particularly for the Asian Summer Monsoon (ASM), and provide support by referring to the literature as well as by comparing with observations (e.g., in lines L32-37; L84-87; L266; L335-336; L439). In doing so, they should provide a stronger connection with the results published in Roozitalab et al., 2024 (cited in the manuscript) as well as to the recent ACCLIP paper from Pan et al., 2024 (https://www.pnas.org/doi/10.1073/pnas.2318716121).*

OK. We have expanded the text at the bottom of Section 3.2 to include more discussion of the findings of Roozitalab et al. (2024). We also now cite and briefly discuss the work of Pan et al. (2024). The new text reads:

"Other recent studies have also highlighted the importance of Asian emissions in contributing to the atmospheric loading of a range of Cl-VSLS. For example, in a global modelling study, Roozitalab et al. (2024) used a 'tagged tracer' approach to show that Asian emissions likely dominate the global $CH_2Cl_2$ and $C_2Cl_4$ distribution. This was the case not only at the surface but also at high altitudes (150 hPa). The same study also analysed measurements of several Cl-VSLS (including DCE) during the ATom campaign and tentatively assigned relatively enhanced NH mid-latitude mole fractions of Cl-VSLS (observed during ATom-1) as being influenced by deep convection associated with the Asian Summer Monsoon. High-altitude aircraft observations from the ACCLIP mission have also revealed that the lower stratospheric abundance of Cl-VSLS above the East Asian monsoon are at least a factor of 2 larger than previously observed in the tropics (Pan et al., 2024)".

We have also added a citation to the Pan et al. paper to the statement around the importance of the ASM in Section 3.3.

*Most importantly, I think it would be a great idea to provide an estimate of the impact of enhanced EDC in the ASM over lower stratospheric ozone during the summer (see specific comment below).*

Please see our response to the specific comment below.

*Given that the inventory considered bi-annual data and you performed a complete simulation for almost 20 years, it would be great to provide the mean rate of growth of EDC both for the surface emission as well as for the SGI and PGI. Those trends values (properly quantified) will be of*

*interest for future reports on VSL influence on stratospheric ozone. In case the trends for the 2002-2020 period differ significantly to the trend during the last 4-5 years, an explicit statement and quantification could be provided.*

OK, this information will be added to the revised manuscript (Section 3.1 for emissions growth and Section 3.2 for SGI/PGI growth).

***Minor Comments:***
*L20: "transport of EDC (or its atmospheric oxidation products)". Is this "or" or "and"?*

*"a*nd/or" is probably the most appropriate. We will amend in the revised manuscript.

*L74: "… at reportedly both urban and background sites …". Please revise text.*

OK. We have removed the word "reportedly".

*L78: Please indicate for which year were estimated the EDC emissions in the refernced study.*

OK. This is based on the measurements from 2013 and 2014. We will amend the sentence in the revised manuscript to include this.

*L133: "However, …". This however seems to indicate discrepancy with previous results, but they all point in the same direction.*

The "however" is used here after the caveat of few estimates being available for Asia.

*L146: "Although the imbalance was small compared to the large production volumes of EDC, …". By how much? At least a percentage number should be provided.*

OK, we will add in the revised manuscript.

*L239: please provide a reference supporting the neglecting of EDC photolysis. Note that other studies cited in this work (Roozitalab et al., 2024) considered photolysis for EDC.*

OK. We now cite Carpenter and Reimann et al. (2014) which states: "*Photolysis of chlorinated SGs is slow and OH oxidation dominates tropospheric loss".*

*L343: I found reasonable to show the median instead of the mean for this case. Just by how much do the mean and the median differ?*

We will add a version of the figure to the Supporting Information with the means.

*L348: I completely agree with the statement and the link with Fig. 3, but I feel that this should also be linked to the spatially heterogeneous source strength shown in Fig. 2a.*

We agree. We have added "elevated *emissions* over East Asia are also a clear feature in Figure 2a".

*L362: "Samples collected at Bachok, where the model captures the shape of the seasonal cycle well, …". I'm really surprised about the large seasonal cycle (both observed and modeled) at this site. Why is this? Is it because the emissions also show a large variability? Is it because OH*

*changes, is it because meteorology? Is it due to influence from continental China? Please extend about this interesting topic !!!*

This is indeed interesting. The observations and model show a similar seasonality but note that the model emissions are non-seasonally varying. This points to emission seasonality as not being the main driver. The explanation is discussed in Oram et al. (2017) and it is dynamical in nature. Briefly, strong north-easterly (NE) winds that form in NH winter transport polluted airmasses from continental East Asia deep into the tropics. The prevailing NE winds may also be strengthened during 'cold surge' events, with the effect of such having been observed for various tracers, including at other sites in Malaysia, e.g. Ashfold et al. (2015) and (2017)
https://doi.org/10.5194/acp-15-3565-2015
https://doi.org/10.1016/j.atmosenv.2017.07.047

In the revised manuscript we will provide some further process level detail to reflect the above and cite these additional studies.

*L414: Given by your methods description, reaction with OH is the only chemical loss in your model. Or does EDC also suffer any type of washout / dry-deposition?*

No deposition of EDC was considered. We will add a note to the Methods section to reflect this in the revised manuscript.

*L432-434: In addition to the referred work for Iodine chemistry, a simplified representation of these ice-recycling reactions has also been performed for bromine and chlorine in Fernandez et al., 2014 (https://doi.org/10.5194/acp-14-13391-2014). The impact of these and other reactions on PGI and ozone loss for the case of bromine was addressed in Fernandez et al., 2021 (https://doi.org/10.1029/2020GL091125), though no estimations was performed for chlorine PGI.*

OK, thank you for pointing this out. We will cite these studies in the revised manuscript and expand the text to reflect the above.

*L475-477: I completely agree with the statement, and suggest that in order to advance in that area, providing an estimation of how much larger is the absolute and/or percentage ozone decrease within the ASM region would be of interest here.*

This is an area that would require a larger dedicated study to examine in detail and would benefit from consideration of additional VSLS species (besides EDC) to see the full potential effect. However, we agree that it is interesting and we will add a figure to the Supporting Information to address this comment.

*L490-493: You should explicitly mention in the conclusion that you used a bottom-up approach.*

OK. We have added "bottom-up" to the following sentence: "Time-varying gridded DCE emission fields were developed **using a bottom-up approach** and then included in the TOMCAT CTM".

*L493-494: "Time-varying gridded EDC emission fields were developed and then included in the TOMCAT CTM." It would be great if you can provide the emission inventory to the community to evaluate it in other models.*

Absolutely. The full gridded emissions will be released in a public repository on acceptance.

***Figures and Tables:***

*Table 4: I understand that in the last row it should say ">60°S" instead of "<"… as southern latitudes are not negative but "South"*

OK, we will amend this.

*Figure 3: would it be possible to show an error bar (spread of data) for the observed data in the vertical profile (panel c)?*

The spread is already indicated with filled grey circles, but they are not so prominent. We will make these a darker shade and more visible in the revised manuscript.

Figure 4: I do not see the shading (sc04 and sc06) but only output for the sc05 results. I think it would be very useful to provide the range here.

The shading is shown but is 'hiding' due the high temporal frequency of data. We will make this more visible in the revised manuscript.

Figure 6: The text mention several times the importance of the high EDC emissions over Asia and the rapid transport due to the ASM, but then the figure highlights the influence during the Antarctic Spring. Wouldn't it be nice to show also delta O3 values for the ASM region during July or August

OK, we agree and will add an additional supplementary figure to show the effect on ozone in this region.